# Response Characteristics of Smoke Detection for Reduction of Unwanted Fire Alarms in Studio-Type Apartments

**Euy-hong Hwang** , **Han-bit Choi** and **Don-mook Choi** *

Department of Equipment System and Fire Protection Engineering, Gachon University, 1342, Seongnam-daero, Sujeong-gu, Seongnam-si 13120, Republic of Korea; dmlghd2@gachon.ac.kr (E.-h.H.); hanbisi970@gachon.ac.kr (H.-b.C.)
* Correspondence: fire@gachon.ac.kr

**Abstract:** Photoelectric smoke detectors (SDs) often emit false alarms in studio-type apartments, where fire prevention is crucial. This study investigates the response characteristics of conventional and analog smoke detection factors to reduce false positives in studio-type apartments. A mock-up was tested based on relevant domestic laws, standards, statistical data, and experimental cases. A simulation of a cooking scenario involving burned food items was conducted, and optical density, particulate matter (PM), and carbon monoxide levels were measured and compared with actual smoke detection at six different locations. The measured values of conventional smoke detectors (CSDs) and analog smoke detectors (ASDs) at these locations were used to derive the activation time of CSDs and ASDs for the entire mock-up space. The results showed that the CSD activated at 7.42 min, while the ASD activated at 11.57 min. PM10.0, CO, and $CO_2$ showed similar activation time trends. The PM10.0, CO, and $CO_2$ concentrations at the time of SD activation were estimated. The findings suggest that a sensor with a consistent coefficient of variation, such as PM10.0 and CO, should be recommended.

**Keywords:** carbon monoxide; particulate matter; photoelectric smoke detector; studio-type apartment; unwanted fire alarm

## 1. Introduction

### 1.1. Research Background and Trends

All fire alarm systems rely on either automatic smoke detection devices or human intervention. In particular, photoelectric smoke detectors (SDs) are devices that use the light scattering method to trigger alarms when smoke enters a detection chamber [1]. These detectors are widely used due to their rapid response time and superior performance compared to heat detectors. In 2015, the installation guidelines for photoelectric smoke detectors were revised in Article 7 of the Automatic Fire Detection System and Visual Warning Device Fire Safety Codes (National Fire Safety Code, NFSC 203), in the Republic of Korea. The revision included the requirement for photoelectric smoke detectors to be placed in living rooms that are used for sleeping, lodging, and hospitalization in specific facilities such as apartments, facilities for the elderly and children, medical facilities, and similar establishments [2].

In the case of a studio-type apartment, it is necessary to install a photoelectric smoke detector in the living room, which is also used as the kitchen. This presents a challenge as the smoke detector lacks adaptability, resulting in frequent unnecessary fire alarms. Photoelectric smoke detectors are therefore essential to improve the reliability of the installation location.

Several studies have investigated the activation characteristics of photoelectric smoke detectors to estimate detection time and compare the response characteristics of smoke detection factors. Quang explored the estimation of detection time in photoelectric smoke

detectors by considering the relationship between light density and temperature, mass light density, and critical velocity [3]. Schifiliti et al. proposed a response model for photosensitive and photoelectric smoke detectors, discussing the concept of unit photosensitivity and the principle of light scattering [4]. Qiyuan et al. examined the possibility of reducing false alarms by developing a multifunctional algorithm using carbon monoxide (CO) and carbon dioxide ($CO_2$) and conducting experiments on false alarms caused by cooking fumes [5]. Zevotek investigated the characteristics of false alarms using fish and bacon [6]. Additionally, Festag et al. discussed the definition of false alarms, their causes and solutions, and recent technological developments to reduce alarms [7]. Choi et al. studied the smoke characteristics of unwanted fire alarms caused by cooking in a studio-type apartment using indoor air quality sensors [8]. However, none of these studies specifically addressed the characteristics of smoke detector installation locations in studio-type apartments.

### 1.2. Research Purpose

The aim of this study was to enhance reliability of smoke detector installation locations, by presenting the unwanted fire alarm scenarios that have the most significant potential to occur in studio-type apartments based on previous research [3–8], and the characteristics of the smoke detection factors were validated through experiments conducted for each scenario.

First, a comprehensive review of the domestic and international literature, standards concerning studio-type apartments, photoelectric smoke detector installation targets, and false fire alarms, along with statistical data and experimental cases based on scenarios of false fire alarms, was conducted to identify existing issues. Second, representative scenarios of false-positive fire alarms that may occur in studio-type apartments were selected, and a large-scale experiment was conducted. Various sensors were installed in an enclosed space, and the specific values of photoelectric smoke detectors at the time of activation were compared and analyzed to assess their suitability in mitigating nuisance alarms. Third, by conducting a comparative analysis of activation characteristics, sensors with similar response tendencies were identified, which enabled the development of an estimation method for smoke detection activation time and the confirmation of estimated values during photoelectric smoke detector activation. Through this process, we propose a sensor that effectively reduces unwanted fire alarms.

## 2. Approach

In this study, installation location and characteristics of smoke detectors and the main causes of unwanted fire alarms are identified. Additionally, using full-scale experiments to investigate unwanted fire safety, scenarios applicable to studio-type apartments were conceived and conclusions drawn.

### 2.1. Studio-Type Apartment and Photoelectric Smoke Detectors Installation Targets

Relevant statistics and regulations were confirmed in the Republic of Korea, where the experiments were conducted.

The National Fire Agency in the Republic of Korea does not collect statistical data on unwanted (false positive) fire alarms through its fire statistical system. Therefore, a pilot fact-finding survey was conducted by the National Fire Agency to establish a data collection system for unwanted fire alarms. The survey on the status of unwanted fire alarms was conducted nationwide from March to May 2018, during which 7849 unwanted fire alarms were reported over three months.

The survey was conducted on all building types, revealing the following results. In terms of building purpose, the largest number of unwanted fire alarms (2267; 29.7%) was reported from amenities, followed by apartments (1639; 21.4%), factories (1222; 16.0%), and facilities for the elderly adults and children (577; 7.5%). Regarding the devices causing unwanted fire alarms, photoelectric smoke detectors accounted for the highest number of cases (2854; 49.9%), followed by heat detectors (1944; 34.0%). The detectors constituted

83.9% (4789 cases) of the total unwanted fire alarm cases. These cases are attributed to detectors, especially smoke detectors, that are installed in places with unsuitable adaptability; this reasoning is confirmed by the law revised in 2015 [9].

According to the Manufactured Product Inspection data collected from March to May 2018 in the Republic of Korea, there were 986,204 smoke detectors, and 4,693,978 heat detectors were found, indicating that there were 4.7 times more heat detectors than smoke detectors [10]. Despite the higher number of heat detectors installed, the fact that smoke detectors accounted for the highest proportion of unwanted fire alarms highlights the severity of the issue caused by smoke detectors. Regarding the causes of unwanted fire alarms, artificial factors accounted for the largest number of cases (1302; 40.6%), followed by managerial factors (1007; 31.4%) and system factors (897; 28%). The Republic of Korea does not have a definition for unwanted fire alarms, and the causes were categorized through an investigation that employed a classification of causes that differed from international standards as follows. Artificial factors encompass heat and smoke generated from cooking, smoke from smoking, vehicle exhaust gases, dust from construction activities, pranks, and human errors. Managerial factors include inadequate cleaning and maintenance of detectors, water leakage through building cracks, and unsuitable environmental conditions surrounding the detectors. System factors refer to errors resulting from aging, construction, and device malfunctions. Cooking-related heat and smoke fall under the category of artificial factors, representing the largest proportion. This suggests that a significant number of unwanted fire alarms are caused by artificial factors in the living spaces of apartments where photoelectric smoke detectors are installed.

According to the regulations, studio-type apartments with a residential area of less than 30 m$^2$ are allowed to have one space (excluding the bathroom and boiler room) where the kitchen must be installed. For studio-type apartments with a residential area between 30 m$^2$ and 50 m$^2$, two spaces (excluding the bathroom and boiler room) are permitted, and the kitchen must be installed in either of those spaces. Moreover, the National Fire Safety Code for Automated Fire Detection Facilities and Visual Alarm Devices (NFSC 203) in the Republic of Korea designates living rooms used for sleeping, lodging, hospitalization, or similar purposes as installation targets for photoelectric smoke detectors. Accordingly, studio-type apartments are considered equivalent to apartments and are thus categorized as installation targets for photoelectric smoke detectors based on these regulations.

Typically, photoelectric smoke detectors are not considered suitable for installation in kitchens. According to Annex 1 of the NFSC 203, fixed-temperature, analog heat, or spark detectors are recommended for kitchen use or areas where smoke is prone to be present. Table 1 provides an overview of recommended smoke detector installation locations in the Republic of Korea. Additionally, areas such as the front room of the kitchen, hallways, and passages adjacent to the kitchen are recognized as locations where a significant amount of smoke can accumulate. In these specific areas, only heat detectors are appropriate for installation. Consequently, in studio-type apartments, which are designated as photoelectric smoke detector installation targets, false alarms may occur due to exposure to cooking fumes in the kitchen. The most significant challenge in studio-type apartments is the coexistence of the kitchen and living room in one space, with adjacent bathrooms and living rooms, which exposes the detectors to potential nuisance sources.

### 2.2. Scenario-Based Experimental Cases on Unwanted Fire Alarms

Figure 1 categorizes the experiments according to unwanted fire alarm scenarios. In the United States, relevant agencies, such as the National Fire Protection Association Fire Protection Research Foundation, Underwriters Laboratory [11], and the National Institute of Standards and Technology [12,13], have conducted Nuisance Alarm Tests involving various cooking activities such as toasting bread, frying bacon, broiling hamburger patties, making grilled cheese sandwiches, vegetable stir-frying and baking pizzas. These tests primarily examined the alarms triggered by cooking by-products (smoke) and water vapor. Notable cases include cooking hamburger patties in an oven or on an electric stove and

deliberately initiating alarms with water vapor from a shower. Additionally, unwanted fire alarm tests were conducted with hamburger patties using standards such as UL 268 and 217. In the UK, the Fire Protection Association [14] conducted experiments where detector activation was tested using water vapor generated by boiling water in a coffee pot. These scenarios were designed to assess whether photoelectric smoke detectors could be activated by substances generated in areas such as kitchens and bathrooms.

**Table 1.** Republic of Korea's representative smoke detector adaptable and impossible places.

| Environmental Conditions | Places | Smoke Detector |
|---|---|---|
| Place of sleeping facility | Hotel room, Motel, Sleeping room, etc. | Applicable |
| Place where dust floats other than smoke | Corridor, Passage, etc. | |
| Kitchen, Other places where smoke usually stays | Kitchen, Galley, Welding Shop | Not Applicable |
| Place where a risk of a great deal inflow of smoke exists | Food distribution room, Kitchen front room, Food storage room in the kitchen, an elevator for transporting food, corridors and corridors around the kitchen, restaurants, etc. | |
| Place where fireworks are exposed as a facility using fire | A glass plant, a place with a furnace, a welding room, a kitchen, a workshop, Kitchen, etc. | |

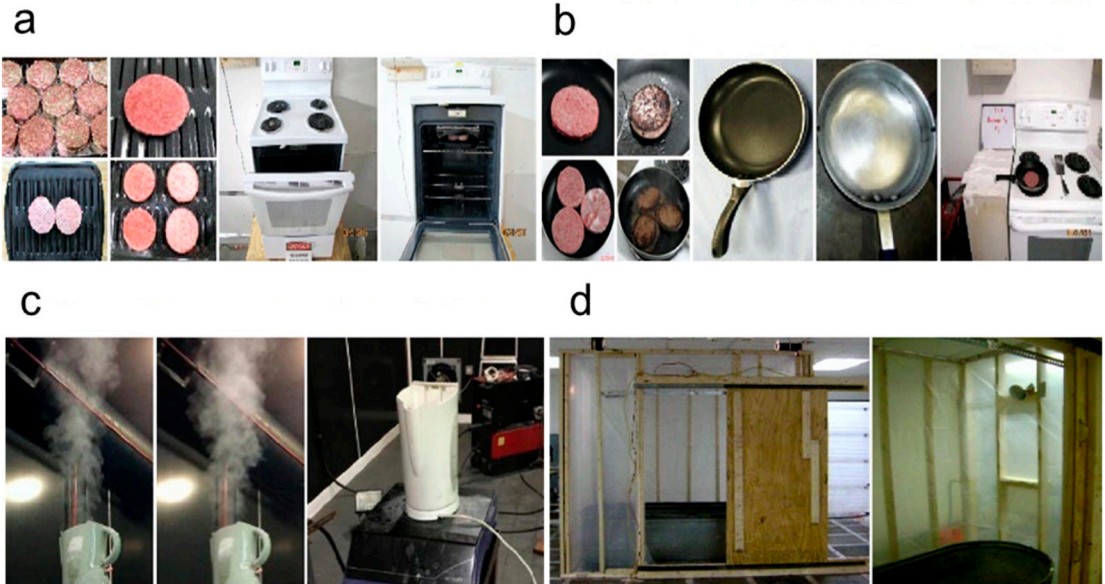

**Figure 1.** Test samples based on alarm scenarios of unwanted fires: (**a**) Grilled hamburger patties in an oven; (**b**) fried hamburger patties in a frying pan; (**c**) boiled water in a coffee pot; (**d**) hot water in a shower room.

## 3. Experiments

### 3.1. Selection of Unwanted Fire Alarm Scenarios

When choosing unwanted fire alarm scenarios, it is important to consider their likelihood of occurrence in real-life situations and the feasibility of reproducing them in experiments. Additionally, as the level of external intervention can affect the results, it is necessary to simplify factors such as artificial and environmental conditions to ensure the accuracy and practicality of the findings. Therefore, scenarios were selected that closely resembled actual situations while minimizing external intervention.

In the chosen scenarios, the photoelectric smoke detectors were triggered by the by-products generated during cooking activities in the kitchen. For the experimental materials, hamburger patties made from ground beef were selected based on international cases, while nubianie (seasoned minced beef), pork belly, and mackerel were chosen based on the domestic situation in the Republic of Korea. Table 2 lists the experimental materials for each specific case.

**Table 2.** Possible and experimental unwanted fire alarm scenarios.

| Case | Scenarios |
| --- | --- |
| 1 | Fried hamburger patties in frying pan |
| 2 | Fried nubianie (seasoned minced beef) in frying pan |
| 3 | Fried pork-belly in frying pan |
| 4 | Fried mackerel in frying pan |

*3.2. Mock-Up Overview*

Figure 2a,b depict the mock-up of a studio-type apartment. According to relevant regulations, a studio-type apartment with an area of less than 30 m$^2$ constitutes a single space. In the experimental mock-up, Room 1 resembled the test site described in ISO 9705 [14], which outlines a full-scale fire test method among the standards of the International Organization for Standardization (ISO). Additionally, a corridor measuring 1.5 m in width and 7 m in length was constructed to examine the activation characteristics of the experimental devices within the corridor (Figure 2b) based on the designated scenarios.

Figure 2c illustrates the location of the experimental devices, which were installed at six positions in the middle of the room and corridor. Devices at positions 1 and 3 were positioned 0.6 m away from the adjacent walls, while the device at position 2 was centrally placed on the ceiling, aligning with the midpoint of the room's length and width. Considering the movement of smoke, the device at position 4 was situated at the intersection of Room 1's centerline and the corridor's centerline in the direction of the source of the fire. Similarly, the devices at positions 5 and 6 were positioned at the intersections of Room 2 and Room 3's centerlines, respectively, with the centerline of the corridor.

The experimental devices are depicted in Figure 2d–g. In particular, Figure 2d illustrates the optical density meter (ODM) used to assess data consistency and measure the obscuration per meter (OPM) of the smoke. The ODM was installed at position 2 in Room 1. At the six designated positions, an analog photoelectric smoke detector (ASD; Figure 2e), a conventional photoelectric smoke detector (CSD; Figure 2f), and an environment sensor (ES; Figure 2g) were installed. Conventional photoelectric smoke detectors (CSDs) operate such that when any of the detectors within a zone detects smoke, the entire zone is triggered into an alarm state. Conversely, ASDs have a unique address for each detector, allowing the fire alarm control panel to precisely identify the exact location of the activated detector. Consequently, if an ASD triggers an alarm, the system can display the specific room or area where the fire was detected. Additionally, ASDs can often provide more nuanced information, such as varying levels of smoke or heat. This capability can assist in determining the severity of the fire. The ES monitors the concentration of particulate matter (PM) based on its diameter, as well as CO and $CO_2$ levels. The ES's PM measurement range included PM1.0 (0.3–1.0 µg/m$^3$), PM2.5 (1.0–2.5 µg/m$^3$), and PM10.0 (2.5–10.0 µg/m$^3$), representing the mass of smoke per unit volume. CO and $CO_2$ were measured in parts per million (ppm). The manufacturer provided the maximum measurement ranges for the ES, which were 1000 µg/m$^3$ for PM (PLANTOWER), 500 ppm for CO (WINSEN), and 10,000 ppm for $CO_2$ (TRUEYES).

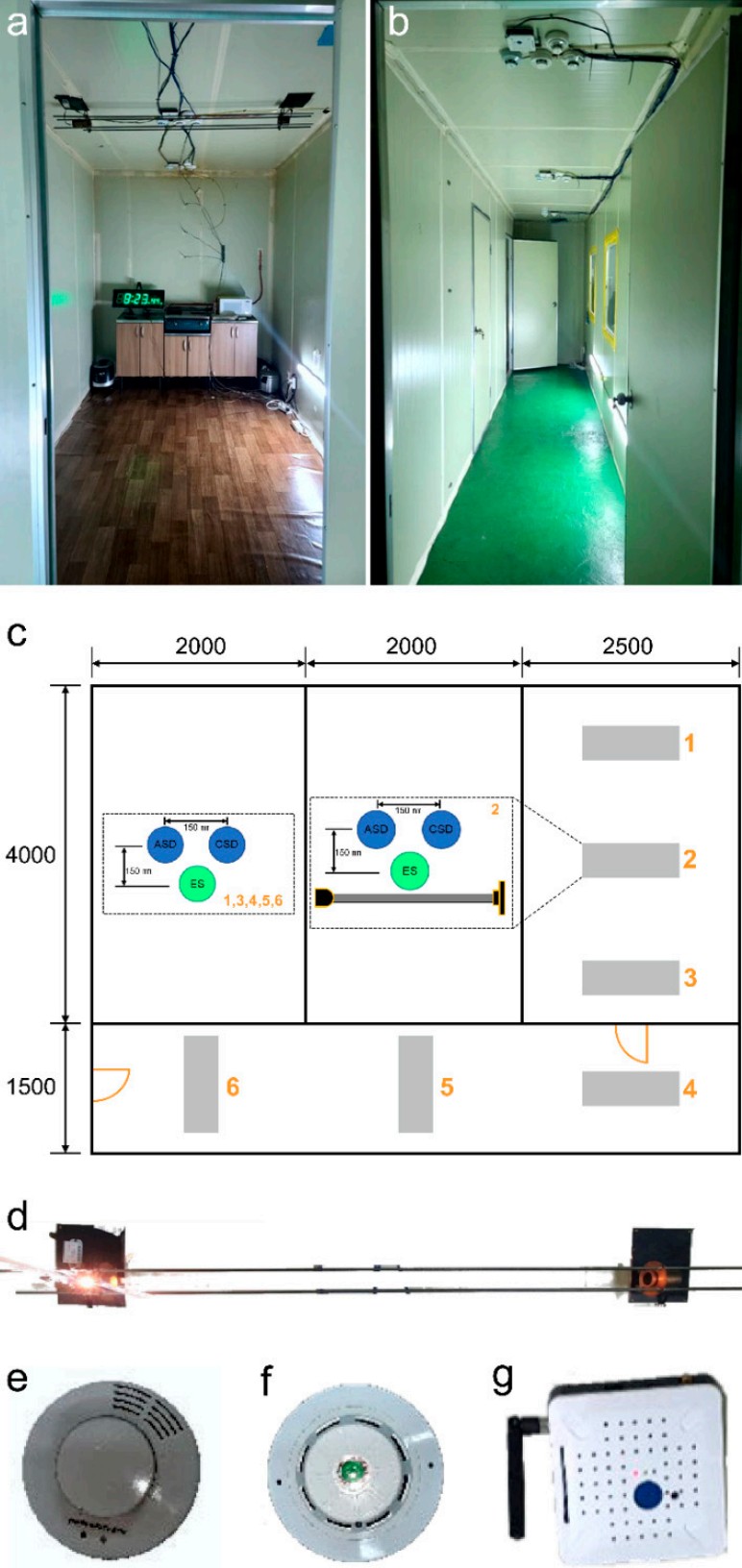

**Figure 2.** Configuration of the mock-up and devices: (**a**) Room 1; (**b**) Corridor; (**c**) Placement of detectors and sensors; (**d**) Optical density meter; (**e**) Analog smoke detector; (**f**) Conventional smoke detector; and (**g**) Environmental sensor.

### 3.3. Experimental Conditions and Method

Regarding the experimental rooms' conditions, the temperature and humidity were set to 23 ± 2 °C and 45 ± 5%. For the corridor, the temperature and humidity were maintained at 20 ± 5 °C and 40 ± 5%, respectively. The air velocity was kept below 0.1 m/s for indoor conditions. Each case was repeated three times to ensure consistency.

Table 3 provides information on the specimens used and outlines the experimental procedure. The initial mass of the samples was 300 ± 10 g, which was determined based on a previous study [12]. All samples were frozen for a minimum of 72 h, and then moved to room temperature immediately prior to the experiment. The frying pan used in the experiment had a diameter of 28 cm; this size was selected as the minimum size that could accommodate all specimens without overlapping. The basic experimental method involved placing the specimens (hamburger patties, nubianie, pork belly, and mackerel) on the frying pan placed on a gas burner for heating. The output of the gas burner was set to its maximum. The gas burner comprised three burners, with a combined maximum gas consumption of 7800 kcal/h (9.1 kW). Two of the burners were small, with a maximum gas consumption of 2180 kcal/h (2.55 kW) each. The large burner used in the experiment had a maximum gas consumption of 3440 kcal/h (4 kW). LPG gas, consisting of 20 kg with a purity of 99.9% or higher propane ($C_3H_8$), was used as the fuel. The flow rate of LPG gas supplied to the gas range through the largest burner port was 2.39 L/min, and the gas flow rate could be controlled.

**Table 3.** Specimen information and the experimental procedure.

| Scenarios | Case 1 | | | Case 2 | | | Case 3 | | | Case 4 | | |
|---|---|---|---|---|---|---|---|---|---|---|---|---|
| Contents | Hamburger patties | | | Nubianie | | | Pork belly | | | Mackerel | | |
| | 1 | 2 | 3 | 1 | 2 | 3 | 1 | 2 | 3 | 1 | 2 | 3 |
| Initial Mass (g) | 300.1 | 299.7 | 305.5 | 303.1 | 305.6 | 309.0 | 304.1 | 305.0 | 305.7 | 305.6 | 308.7 | 309.4 |
| Final Mass (g) | 128.2 | 125.4 | 132.9 | 161.4 | 149.6 | 152.2 | 78.7 | 93.2 | 64.0 | 140.2 | 143.6 | 145.8 |
| Cooking Oil Mass (g) | 29.9 | 30.1 | 32.8 | 29.8 | 28.5 | 30.0 | Not used | Not used | Not used | 30.7 | 30.2 | 31.4 |
| Start |  | | |  | | |  | | |  | | |
| Turn over (08:00) |  | | |  | | |  | | |  | | |
| Finish |  | | |  | | |  | | |  | | |

The start of the experiment was defined as the moment when the material was placed on the preheated frying pan, reaching a temperature of 180 ± 20 °C. To prevent the materials from sticking to the pan, approximately 30 ± 3 g of cooking oil was added, except for Case 3 (pork belly). Following 8.00 min, the materials were turned over once to minimize external intervention. The experiment was defined as complete when all smoke detectors were activated.

## 4. Results and Discussion

### 4.1. OPM Measurement

Figure 3a illustrates the results of the OPM measurements, displaying the average OPM values obtained from the three repeated measurements for Cases 1 to 4. Figure 3b is the three repeated measurements OPM's standard deviation by bar. Deviation, despite variations in the overall OPM levels and due to factors, such as material type and position, resulted in similar patterns when examining the tendencies across the three experiments conducted for each case. This consistency may have been influenced by the surface temperature of the frying pan for each case, as shown in Figure 4, as well as the relative humidity specific to each case, as depicted in Figure 5. Figures 4b and 5b represent the standard deviation of all experiments per case.

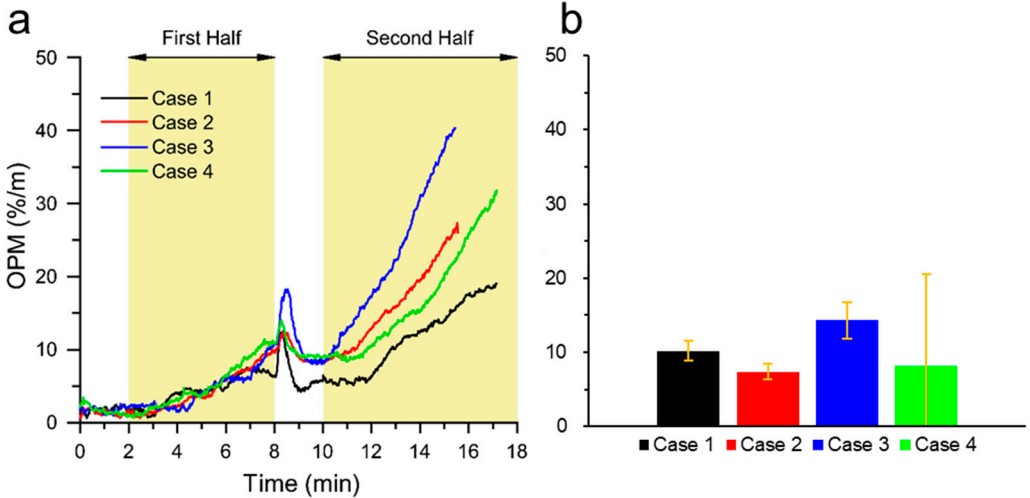

**Figure 3.** Obscuration over time of the obscuration per meter: (**a**) Data; (**b**) Standard deviation.

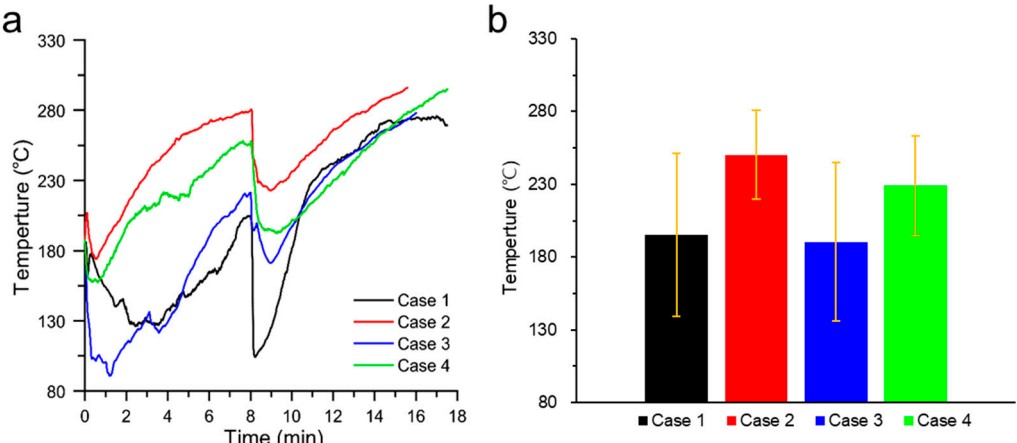

**Figure 4.** Surface temperature over time of the frying pan: (**a**) Data; (**b**) Standard deviation.

A notable common pattern was observed in the experimental results. The first pattern was characterized by a point of inflection where the temperature slightly decreased, and the relative humidity slightly increased after adding the material at the initial preheating temperature of $180 \pm 20\ °C$. This occurrence can be attributed to the generation of fine smoke, including water vapor, when the ice attached to the frozen material comes into contact with the hot surface of the frying pan.

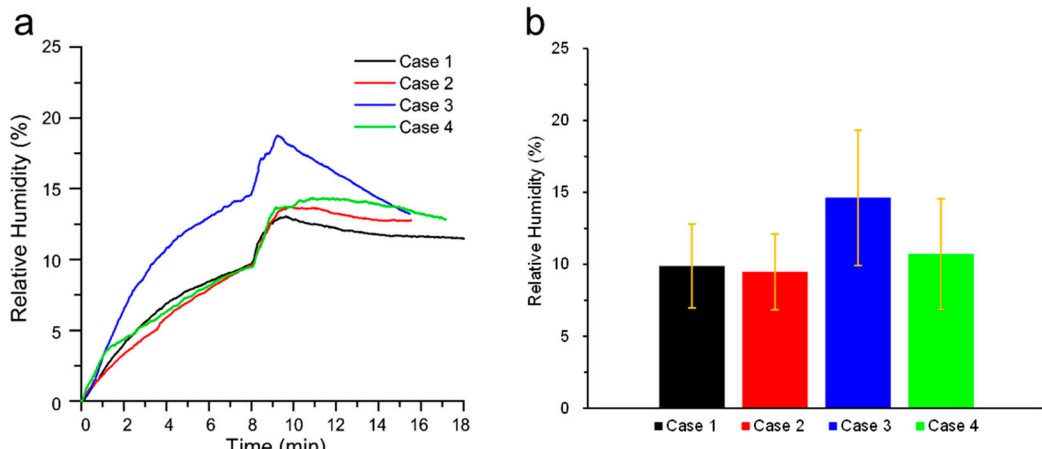

**Figure 5.** Humidity over time in the mock-up devices: (**a**) Data; (**b**) Standard deviation.

The second pattern also displayed a point of inflection when the material was flipped over after being left in the pan for 8.00 min. The peak time was observed to be approximately $8.33 \pm 0.33$ min. The reason for this pattern is that the temperature of the pan increased beyond the initial preheating level after 8.00 min. Flipping the material at this moment resulted in a higher OPM due to the instantaneous release of water vapor, as the condensed moisture on the surface of the uncarbonized material, which was at a relatively low temperature, came into contact with the hot pan surface.

The third observed pattern revealed similarities between the rising trends during the initial material addition (referred to as the first buildup) and when the material was flipped over (referred to as the second buildup). The experimental results indicated that the first buildup began at around $2.00 \pm 1.00$ min and continued until the moment of flipping over the material. Subsequently, the second buildup started at approximately $10.00 \pm 1.00$ min, which was approximately $2.00 \pm 1.00$ min after flipping over the material, and persisted until the end. These findings suggest that the moisture within the material rapidly dissipated within the initial $2.00 \pm 1.00$ min. Consequently, in experiments involving frozen cooked products, the actual burning of the material takes place in earnest after the surface, and interior moisture of the material evaporates within approximately $2.00 \pm 1.00$ min. Moreover, the minimum required time for one side to be burnt is 8.00 min.

### 4.2. Estimation of the Activation Time of the Photoelectric Smoke Detector

The three repeated experiments showed that the ASD was activated when the OPM reached and sustained around 15%/m for approximately 0.08 min. As the CSDs operates through light scattering, it does not provide numerical OPM values. Therefore, in this case, the OPM value of the adjacent ASD at the time of activation was used as a reference. When the CSD was activated, the OPM value of the neighboring ASD ranged from 5 to 12%/m. These results enabled us to determine that the CSD activates at an average OPM value of 8.5%/m. To estimate the activation times of the CSD and ASD, the average activation time of the photoelectric smoke detector for each case and position, as well as the OPM change rate of the photoelectric smoke detectors, were examined.

Table 4 presents the activation times of the photoelectric smoke detectors at each position for each case. Overall, the CSDs tended to activate earlier than the ASDs. Among the detectors that were activated over time, only two CSDs were activated between 0.00 and 0.33 min upon adding the material, while four CSDs were activated at $7.00 \pm 1.00$ min during the first buildup process. During the turning over of the material, between 8.00 and 8.67 min, three CSDs and four ASDs were activated. At $14.00 \pm 2.00$ min during the second buildup process, most of the photoelectric smoke detectors were activated, including 15 CSDs and 13 ASDs. After 16.00 min, 8.00 min after the material flip, the remaining seven ASDs were activated.

**Table 4.** Activating time of photoelectric smoke detectors according to the point of each case.

| OPM (%/m) | Time (min) | Case | | | |
|---|---|---|---|---|---|
| | | 1 | 2 | 3 | 4 |
| Conventional Photoelectric Smoke Detectors (8.5) | Point_1 | 0.23 | 0.12 | 6.27 | 7.12 |
| | Point_2 | 13.63 | 8.52 | 13.58 | 14.10 |
| | Point_3 | 13.95 | 12.90 | 13.88 | 15.60 |
| | Point_4 | 7.27 | 8.45 | 8.63 | 7.98 |
| | Point_5 | 14.32 | 13.17 | 12.42 | 15.17 |
| | Point_6 | 15.30 | 13.40 | 12.78 | 14.82 |
| Analog Photoelectric Smoke Detector (15) | Point_1 | 8.10 | 8.12 | 8.10 | 8.13 |
| | Point_2 | 15.92 | 13.83 | 15.20 | 16.17 |
| | Point_3 | 17.62 | 14.27 | 15.47 | 16.78 |
| | Point_4 | 14.97 | 13.40 | 12.47 | 14.67 |
| | Point_5 | 17.22 | 15.15 | 13.75 | 16.97 |
| | Point_6 | 18.12 | 15.38 | 13.93 | 17.58 |

The results suggest that unwanted fire alarms may occur when a rapid fluctuation in the amount of water vapor exists due to temperature differences between the pan and the material, such as during the addition and flipping of the material. Additionally, unwanted fire alarms can occur when a sufficient amount of cooking by-products, capable of activating the photoelectric smoke detectors, accumulate after approximately $8.00 \pm 1.00$ min from the time of material addition or flipping, which initiates the buildup process. Note that, these results, which involve artificial interventions, were excluded when considering the OPM gradient of the photoelectric smoke detectors, as the photoelectric smoke detectors were determined to be activated by water vapor rather than cooking by-products.

The experiments showed that the six photoelectric smoke detectors did not activate sequentially. For a photoelectric smoke detector to activate, cooking by-products need to enter the internal chamber of the photoelectric smoke detector and cause light scattering, triggering a signal. The smoke generated during the cooking scenario has a lower temperature than the actual fire, resulting in lower thermal buoyancy and a relatively small updraft. Consequently, the inflow of smoke exhibited irregular characteristics, regardless of the location.

The ASD located at point 2, as shown in Figure 6a, exhibited similar behavior to the average ASD of all points depicted in Figure 6b. Figure 6c shows all of the experiment's standard deviation case by case. This implies that the overall smoke generation throughout the mock-up space is comparable to the ASD value measured at point 2. Consequently, based on the average results of the OPM gradient of the smoke detectors and their activation times, an equation for the activation time of the smoke detectors was derived.

To minimize errors when calculating the overall gradient, the OPM gradient was determined separately for the first and second half of the experiments. Table 5 presents the OPM gradients of the smoke detectors. By analyzing the OPM gradient during the period from the first buildup until the moment of material flipping (referred to as the first half), and from the second buildup until the occurrence of maximum OPM (referred to as the second half), the gradient for the entire experiment was predicted.

Two minutes after the ingredients were added and 2 min after flipping them over, the moisture within the material rapidly dissipated. Therefore, the onset of the gradient for the first and second halves occurred at 2.00 and 10.00 min, respectively. The gradients for the first half were 0.804%/m·min for Case 1, 0.948%/m·min for Case 2, 0.558%/m·min for Case 3, and 0.840%/m·min for Case 4. The gradients for the second half were 2.052%/m·min for Case 1, 2.772%/m·min for Case 2, 2.526%/m·min for Case 3, and 1.914%/m·min for Case 4. These findings indicate that the gradients in the second half were higher than those in the first half. This could be attributed to the rapid increase in smoke,

including cooking by-products during the second half, as the rise in temperature enhanced efficiency and carbonization with decreasing moisture in the material. The overall gradient can be expressed as the average sum of the gradients for the first and second halves, as derived earlier. Considering the relationship between the gradient and activation time measured in OPM for the smoke detectors, an equation was proposed to estimate the activation time at CSD 8.5%/m and ASD 15%/m.

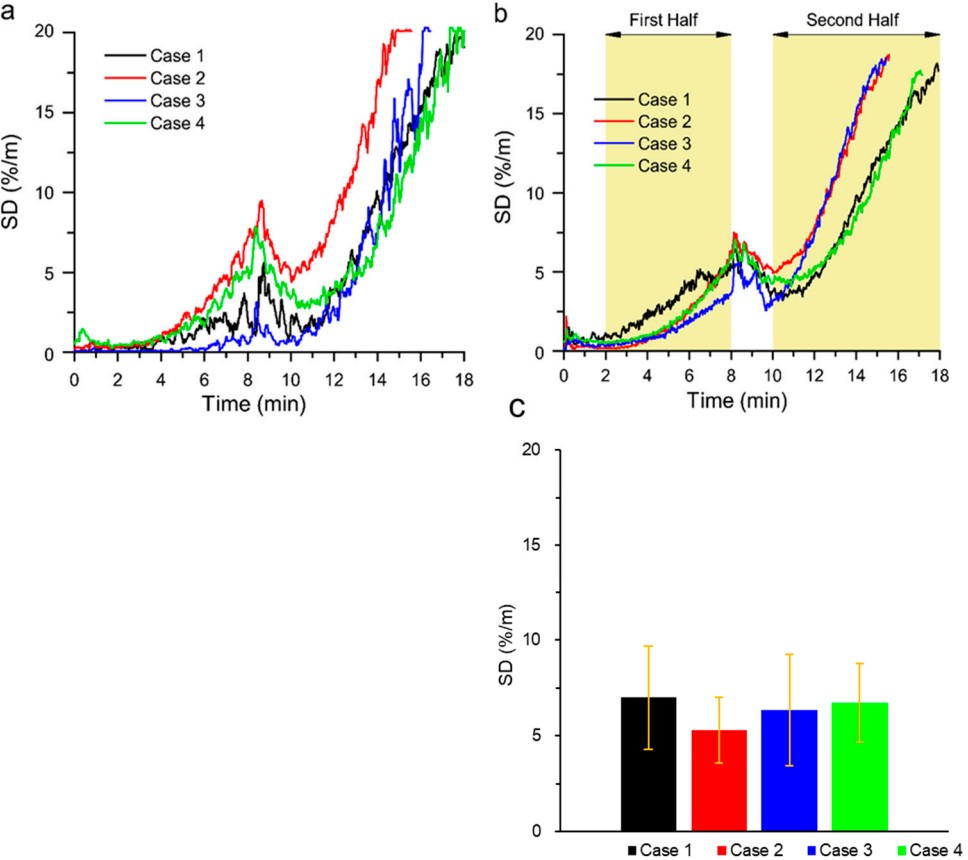

**Figure 6.** Obscuration over time of the analog smoke detector: (**a**) Point 2; (**b**) Average; (**c**) Standard deviation.

**Table 5.** OPM gradient of photoelectric smoke detectors.

| OPM_Gradient (%/m·min) | Case | | | |
|---|---|---|---|---|
| | 1 | 2 | 3 | 4 |
| First half * | 0.804 | 0.948 | 0.558 | 0.840 |
| Second half ** | 2.052 | 2.772 | 2.844 | 1.914 |

* From 2–8 min / ** From 10 min to the occurrence of maximum obscuration.

Table 6 presents the estimation activation times of CSD and ASD. The formulas used to calculate the activation times of the photoelectric smoke detectors are shown below. These equations were derived to estimate the operating times of smoke detectors that activate in random order. The response level of the ASDs can be adjusted by modifying each parameter (gradient). Therefore, the equation was derived based on the gradient to provide a general representation of the operating sequence of the detectors. The activation time of the smoke detector is influenced by the resistance encountered at the entrance, which is known as entry lag. This entry lag can be expressed using the following equation:

$$\frac{dOD_i}{dt} = \frac{1}{\tau}(OD_o - OD_i)\ s^{-1} \cdot m^{-1} \tag{1}$$

where $OD_o \, (\text{m}^{-1})$ is the detector's external optical density (OD) at the time of response, $OD_i (\text{m}^{-1})$ is the internal OD required to enable the detector response, and $OD\left(\text{m}^{-1}\right)$ is the OD per unit length. Thus, Equation (1) was expressed by a gradient, as shown in Equation (2) [5]. In addition to smoke characteristics and the detector's operating mechanism, entry resistance, which is the ability to get the smoke into the chamber, affects the response of the unit. For spot-type photoelectric smoke detectors, entry resistance is caused by bug screens, chamber design, and the detector's aerodynamic characteristics. In a scenario where the OD at the detector location increases with time, the OD inside the detector chamber will always be less than that outside the detector chamber. Similarly, if a detector is placed in a smoke stream having a constant OD, there will be a time delay before the OD inside the chamber approaches that which is outside the detector. If the time constant and rate of change of OD outside the detector are constant, this equation can be solved. Since we installed the same detector, this equation was applied. The relationship between the $OD$ (m$^{-1}$) and OPM can be expressed as Equation (3), according to UL 217 Annex B.

$$\frac{(OD_o - OD_i)}{\tau} = OD_{Gradient}, \tag{2}$$

$$OD = -log_{10}\left(\frac{100 - OPM}{100}\right). \tag{3}$$

**Table 6.** Estimation of photoelectric smoke detector's activation times.

| Type | Time$_{\text{SD}}$ (min) | | | |
| --- | --- | --- | --- | --- |
| | Case 1 | Case 2 | Case 3 | Case 4 |
| CSD | 7.95 | 6.57 | 7.00 | 8.17 |
| ASD | 12.50 | 10.06 | 10.82 | 12.89 |

Here, dOD$_i$ represents the optical density during detector operation, as shown in Equation (4), while dt corresponds to the operating time of the detector minus the time before flipping the sample (2 min), since the amount detected before flipping is close to zero and there is no random intervention, as shown in Equation (5). To obtain the average value, Equation (6) is applied by separating the first half (before input) and the second half (after input) based on flipping, which can be expressed as shown in Equation (7). Furthermore, this can be expanded as Equation (8) to derive Equation (9).

$$dOD_i = SD_{OPM}, \tag{4}$$

$$dt = Time_{SD} - 2, \tag{5}$$

$$OPM_{Gradient} = \frac{OG_f + OG_s}{2}, \tag{6}$$

$$Time_{SD} - 2 = \frac{SD_{OPM}}{OPM_{Gradient}}, \tag{7}$$

$$Time_{SD} - 2 = \frac{2}{OG_f + OG_s} \cdot (SD_{OPM}), \tag{8}$$

$$Time_{SD} = 2\left(1 + \frac{SD_{OPM}}{OG_f + OG_S}\right), \tag{9}$$

where $\tau$(s) is the time constant of the detector (time point when the detector is activated).

$Time_{SD}$ represents the activation time of the photoelectric smoke detector. ($Time_{SD} - 2$) indicates the activation time of the photoelectric smoke detector starting 2 min after the beginning of the experiment. $OG_f$ refers to the gradient of the first half, $OG_s$ represents the gradient of the second half, and $SD_{OPM}$ denotes the OPM value required for the smoke detector activation. Assuming that the internal light sensitivity of the detectors is equal, given that the initial experiment conditions are identical, the anticipated $SD_{OPM}$ can be estimated by multiplying $OPM_{Gradient}$, derived from the experiment, by $Time_{SD}$.

The OPM value triggering photoelectric smoke detector activation was obtained by substituting 8.5%/m for CSD and 15%/m for ASD. The results indicated that the activation time for CSD was 7.95 min for Case 1, 6.57 min for Case 2, 7.00 min for Case 3, and 8.17 min for Case 4. In contrast, the activation time for ASD was 12.50 min for Case 1, 10.06 min for Case 2, 10.82 min for Case 3, and 12.89 min for Case 4.

### 4.3. Measurements and Gradient of Environmental Sensor

The gradients of OPM, PM, and carbon oxides were compared to assess the ES factors that can enhance the adaptability of photoelectric smoke detectors to unwanted fire alarms. The gradients were determined based on the average ES values at 6 points.

First, the gradient of PM measured using the ES was analyzed. PM1.0, PM2.5, and PM10.0, which have different particle diameters, exhibited variations depending on the scenario. Regardless of fire or unwanted fire alarms, PM10.0 showed an increasing trend, while PM1.0 demonstrated a decreasing trend as the visible smoke levels escalated. PM2.5 fell between these two values. Figure 7a–c illustrate the gradients of PM1.0, PM2.5, and PM10.0, respectively. In all experiments, before flipping the samples, PM1.0, PM2.5, and PM10.0 exhibited higher values in that order. After flipping the samples, a significant amount of smoke was generated, resulting in higher values in the order of PM10.0, PM2.5, and PM1.0. Figure 7d–f exhibit the standard deviation of PMs.

The sensor characteristics showed that PM1.0 increased due to moisture from the burnt material, as indicated by the corresponding increase in relative humidity. Notably, both PM1.0 and relative humidity rapidly increased between 0.00 and 2.00 min and between 8.00 and 10.00 min. However, they subsequently showed a rapid decrease. Moreover, PM2.5 displayed a slightly faster increase than the OPM gradient at the time of material addition. However, it tended to decrease during the second buildup, which involved carbonization. Therefore, applying PM2.5 as a characteristic factor for activating unwanted fire alarms in photoelectric smoke detectors is also impractical. In contrast, PM10.0 exhibited a pattern similar to the OPM gradient, suggesting that it could be employed as a characteristic factor for activating unwanted fire alarms in photoelectric smoke detectors.

Subsequently, the carbon oxides CO and $CO_2$ were examined as measured by the ES. While CO traditionally represents incomplete combustion products and $CO_2$ represents complete combustion products, both were measured in this study due to their potential occurrence during cooking. Although CO is typically measured in ppm and $CO_2$ in % concentration, both were measured in ppm for faster gradient measurement. Figure 8a,c illustrate the time-dependent generation of CO and $CO_2$, respectively, and Figure 8b,d exhibit the standard deviation of CO and $CO_2$. While the gradients of the carbon oxides slightly differed from the OPM gradient, both CO and $CO_2$ exhibited consistent increases. This suggests their suitability as characteristic factors for activating unwanted fire alarms in photoelectric smoke detectors. Consequently, using only one factor for verifying the respond characteristics and reducing unwanted fire alarms in photoelectric smoke detectors is challenging.

Table 7 lists environmental sensor gradients (EG) of the three identified characteristic factors applicable to unwanted fire alarms based on the experimental results discussed above. By applying these gradients, the $ES_{data}$ can be estimated.

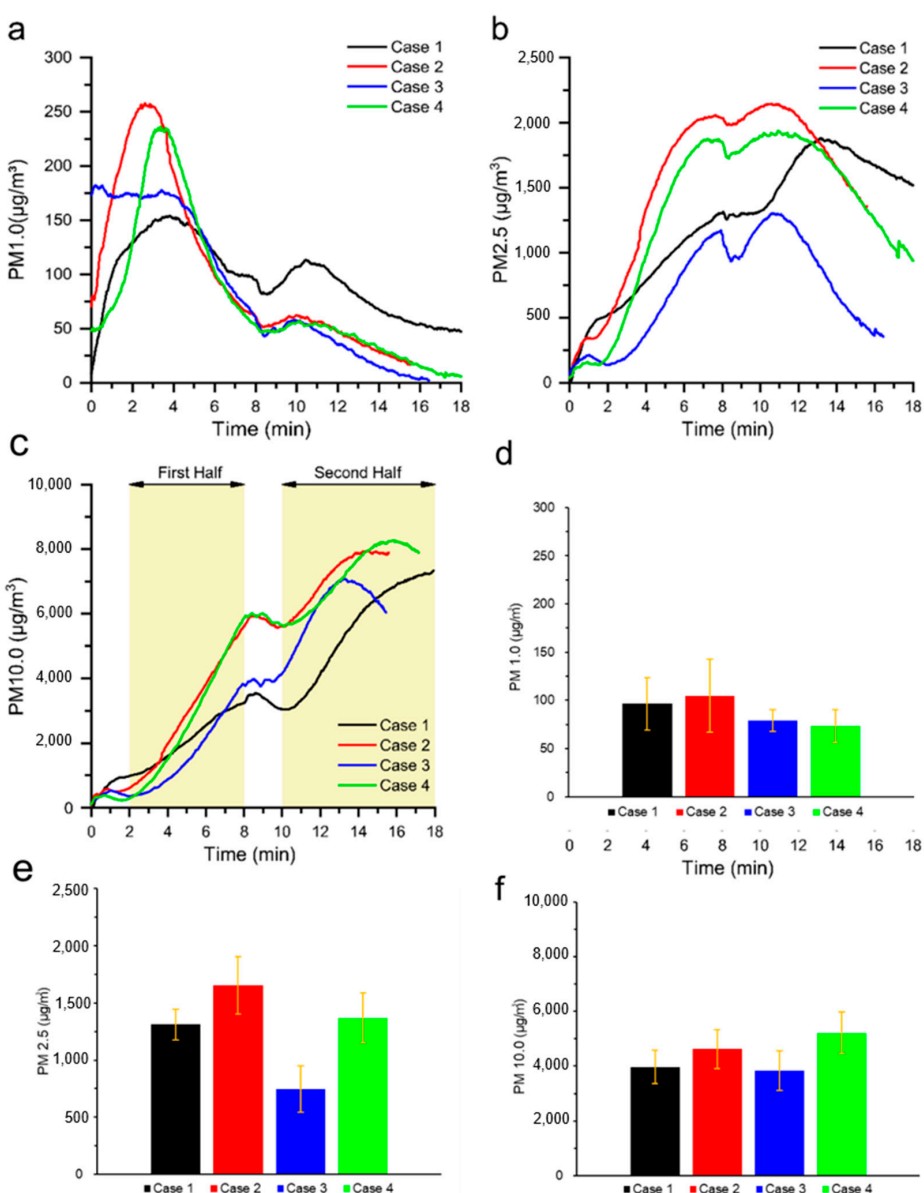

**Figure 7.** Microgram per cubic meter over time of particulate matters: (**a**) PM1.0; (**b**) PM2.5; (**c**) PM10.0; (**d**) Standard deviation of PM1.0; (**e**) Standard deviation of PM2.5; (**f**) Standard deviation of PM10.0.

**Table 7.** Gradient of environment sensors.

| ES_Gradient | Case | Particulate Matters ($\mu g/m^3 \cdot min$) | Carbon Oxides (ppm/min) | |
|---|---|---|---|---|
| | | PM10.0 | CO | $CO_2$ |
| First half * | 1 | 424.398 | 0.168 | 131.136 |
| | 2 | 891.240 | 0.312 | 139.974 |
| | 3 | 617.220 | 0.150 | 134.928 |
| | 4 | 983.340 | 0.210 | 113.088 |
| Second half ** | 1 | 583.164 | 0.222 | 104.610 |
| | 2 | 474.408 | 0.444 | 126.798 |
| | 3 | 983.760 | 0.246 | 109.380 |
| | 4 | 543.708 | 0.120 | 110.622 |

* From 2–8 min/** From 10 min to the occurrence of maximum obscuration.

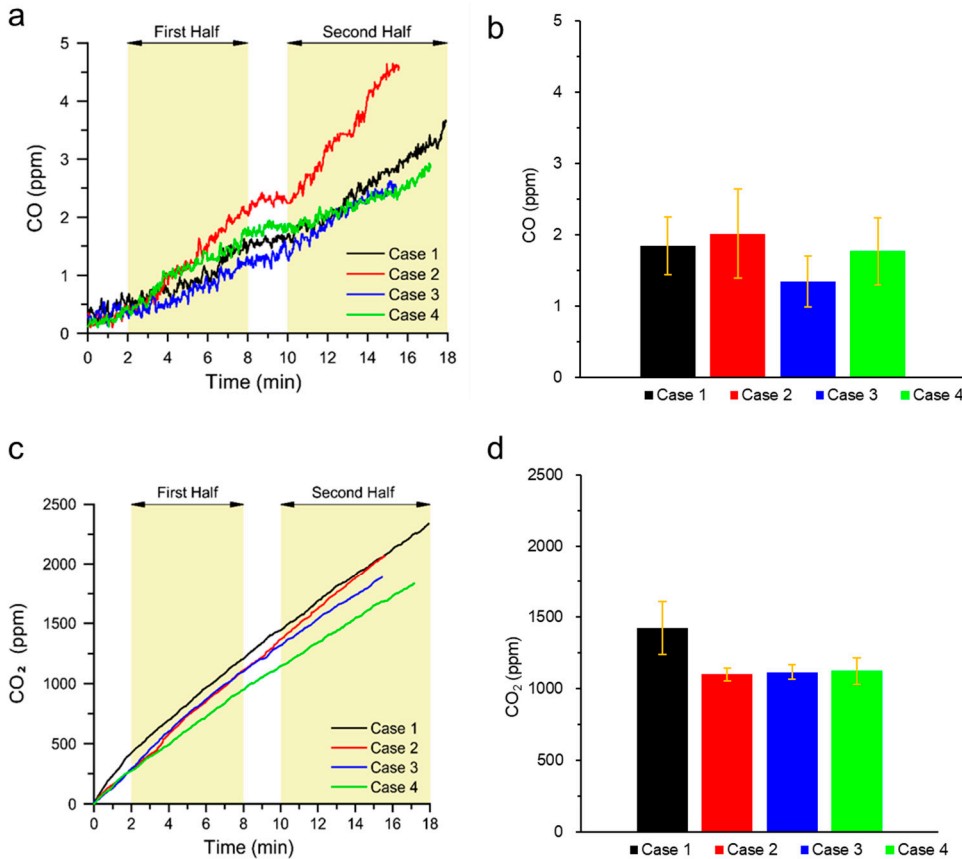

**Figure 8.** Percent per million over time of carbon oxides: (**a**) CO; (**b**) Standard deviation of CO; (**c**) $CO_2$; (**d**) Standard deviation of $CO_2$.

These results differ from the slope values inferred using the unwanted fire alarm judgment area values proposed in previous studies, such as PM10.0 (295.7 μg/m³ min), CO (0.615 ppm/min), and $CO_2$ (23.35 ppm/min) [15]. This difference is attributed to variations in the experimental environment, such as the height of the space and the presence of personnel intervention during sample insertion and turning.

By examining the gradients of PM10.0, CO, and $CO_2$ as characteristic factors for unwanted fire alarms, an equation was derived to estimate the environmental sensor data ($ES_{data}$) at the time of photoelectric smoke detector activation. Based on these findings, Equation (11) illustrates the method for deriving the ES value, which exhibits a characteristic trend when the sensor activates in a cooking scenario.

$$Time_{SD} = 2\left(1 + \frac{ES_{data}}{EG_f + EG_S}\right), \tag{10}$$

$$ES_{data} = \left(EG_f + EG_s\right) * \left(\frac{Time_{SD}}{2} - 1\right). \tag{11}$$

where $ES_{data}$ is the environmental sensor data, $EG_f$ is environmental sensor gradient of the first half and $EG_s$ is environmental sensor gradient of the second half.

The activation time of the photoelectric smoke detectors, PM10.0, CO concentration, and $CO_2$ concentration values measured with ES, along with their standard deviations, are presented in Table 8. The activation time of the CSD was 7.42 ± 0.76 min. Simultaneously, the PM10.0 value was 3707.85 ± 804.68 μg/m³, and the CO and $CO_2$ concentrations were 1.23 ± 0.34 ppm and 652.96 ± 42.97 ppm, respectively. The activation time of the ASD was determined to be 11.57 ± 1.35 min. Simultaneously, the PM10.0 value was

$6542.09 \pm 1421.03$ μg/m$^3$, and the CO and CO$_2$ concentrations were $2.16 \pm 0.61$ ppm and $1152.07 \pm 87.99$ ppm, respectively.

**Table 8.** Estimation of value and STDEV (Standard deviation) at photoelectric smoke detector activation time.

| Estimation Value | Case | Time$_{SD}$ (min) | PM10.0 (μg/m$^3$) | CO (ppm) | CO$_2$ (ppm) |
|---|---|---|---|---|---|
| CSD_ Activation | Case 1 | 7.95 | 2997.50 | 1.16 | 701.34 |
| | Case 2 | 6.57 | 3120.51 | 1.73 | 609.57 |
| | Case 3 | 7.00 | 4002.45 | 0.99 | 610.77 |
| | Case 4 | 8.17 | 4710.94 | 1.02 | 690.15 |
| | Average | 7.42 | 3707.85 | 1.23 | 652.96 |
| | STDEV. | 0.76 | 804.68 | 0.34 | 42.97 |
| | Maximum (Avg + Std) | 8.18 | 4512.53 | 1.57 | 695.93 |
| | Minimum (Avg − Std) | 6.66 | 2903.17 | 0.89 | 609.99 |
| | Coefficient of Variation | | 0.22 | 0.28 | 0.07 |
| ASD_ Activation | Case 1 | 12.50 | 5289.70 | 2.05 | 1237.67 |
| | Case 2 | 10.06 | 5503.56 | 3.05 | 1075.09 |
| | Case 3 | 10.82 | 7060.32 | 1.75 | 1077.40 |
| | Case 4 | 12.89 | 8314.78 | 1.80 | 1218.10 |
| | Average | 11.57 | 6542.09 | 2.16 | 1152.07 |
| | STDEV. | 1.35 | 1421.03 | 0.61 | 87.99 |
| | Maximum (Avg + Std) | 12.92 | 7963.12 | 2.77 | 1240.06 |
| | Minimum (Avg − Std) | 10.22 | 5121.06 | 1.55 | 1064.08 |
| | Coefficient of Variation | | 0.22 | 0.28 | 0.09 |

The applicability of PM10.0, CO, and CO$_2$ as sensors to supplement the unwanted fire alarms of photoelectric smoke detectors was examined by assessing the degree of data dispersion from the average. The coefficient of variation, obtained by dividing the standard deviation by the mean, was used to measure the degree of dispersion. PM10.0 and CO exhibited the same coefficients of variation (0.22 and 0.28, respectively) for both CSD and ASD, indicating their suitability as characteristic factors for unwanted fire alarm activations. The discrepancy in values compared to previous studies (PM10.0: 5914 μg/m$^3$ and CO concentration: 12.3 ppm) is likely due to environmental differences, such as the shape and height of the space and the device installation method [16]. However, CO$_2$ showed different coefficients of variation for CSD (0.07) and ASD (0.09), indicating that CO$_2$ is not suitable as a characteristic factor. This could be attributed to the influence of human intervention, as mentioned earlier. The coefficient of variation, representing the relative standard deviation, is a unitless measure that allows for comparing data sets with different units of measurement.

### 4.4. Discussion of Unwanted Fire Alarm Reduction Sensors and Limitations

The response characteristics of smoke detectors were investigated through experiments and analysis. This research suggests that unwanted fire alarms may occur when rapid fluctuations in the amount of vapor exist due to temperature differences between the pan and the sample, such as when adding or flipping the material. Additionally, the experiments revealed that six photoelectric smoke detectors did not operate sequentially. This is because the temperature is lower than that of an actual fire, resulting in low thermal buoyancy and relatively small updrafts. Since this factor interferes with improving reliability by

understanding the characteristics of smoke detectors, we proposed equations that can estimate the activating time of smoke detector by considering the activation time.

Additionally, smoke detection characteristics were identified through various environmental sensors. The PM10.0 and CO sensors have been identified as effective in reducing unwanted fire alarms based on the findings. The activation times of 7.42 ± 0.76 min (CSD) and 11.57 ± 1.35 min (ASD) were inferred using equations and the photoelectric smoke detector gradient.

Using these research results, the location where unwanted fire alarms occur in a studio-type apartment due to cooking by-products can be predicted. Furthermore, the results can be used to install smoke detectors in the most suitable places to reduce unwanted fire alarms.

However, note that a slight discrepancy exists between Y.G. Choi's study and the standards (ISO 7240-6, UL 268) in determining the gradient and the value of CO and PM10.0 as sensors to help reduce unwanted fire alarms. Although, this discrepancy is not unreasonable, as similar results have been observed [16–18]. In particular, the value of CO obtained in this study is significantly lower than the gradient and value in a fire situation. Nevertheless, this value is still considered suitable value because it shows similarity to the gradient and value proposed in the cooking disturbance smoke alarm test of UL 268, which is an international standard. Although domestic and foreign research on this specific aspect is scarce, PM10.0 is considered effective because it falls within the particle size range (0.3–10.0 μm) of smoke that can be detected by a photoelectric smoke detector [5].

Note that the gradient of each sensor may vary depending on the manufacturer and device; thus, additional verifications are necessary for future work. Additionally, the characteristics may be slightly different in spaces other than studio apartments. Therefore, further validation is needed for future work. We are conducting research in various spaces, such as stairs, and are considering a variety of materials that may occur in addition to the four scenarios.

## 5. Conclusions

This study determines the characteristics of smoke detection through experiments conducted for multiple scenarios to reduce unwanted fire alarms of photoelectric smoke detectors in a studio-type apartment setting. The main findings of this study are as follows:

First, the estimated activation time of photoelectric smoke detectors based on the OPM of the photoelectric smoke detector and ODM; the average activation time was determined as 7.42 ± 0.76 min for CSD and 11.57 ± 1.35 min for ASD.

Second, an equation was introduced to derive the ES values corresponding to the OPM threshold for photoelectric smoke detector activation. This equation enabled to compute the concentrations of PM10.0, CO, and $CO_2$, which were estimated as 3707.85 ± 804.68 μg/m$^3$, 1.23 ± 0.34 ppm, and 652.96 ± 42.97 ppm, respectively, when CSD was activated. Similarly, the concentrations of PM10.0, CO, and $CO_2$ were estimated as 6542.09 ± 1421.03 μg/m$^3$, 2.16 ± 0.61 ppm, and 1152.07 ± 87.99 ppm, respectively, when ASD was activated.

Third, PM10.0, CO, and $CO_2$ in the ES exhibited gradients similar to those of photoelectric smoke detectors in non-fire situations. Comparing their coefficients of variation indicated that $CO_2$ had low applicability, while PM10.0 and CO showed potential as sensors for supplementing unwanted fire alarms.

**Author Contributions:** Conceptualization: D.-m.C. and E.-h.H.; methodology: D.-m.C. and E.-h.H.; validation: D.-m.C., E.-h.H. and H.-b.C.; formal analysis: E.-h.H.; investigation: E.-h.H.; data curation: H.-b.C.; writing—original draft preparation: E.-h.H.; writing—review and editing: E.-h.H. and H.-b.C.; visualization: H.-b.C.; supervision: D.-m.C.; project administration: D.-m.C. All authors have read and agreed to the published version of the manuscript.

**Funding:** This research received no external funding.

**Institutional Review Board Statement:** Not applicable.

**Informed Consent Statement:** Not applicable.

**Data Availability Statement:** All data generated or analyzed during this study are included in this. Published article.

**Conflicts of Interest:** The authors declare no conflict of interest. The funders had no role in the design of the study; in the collection, analyses, or interpretation of data; in the writing of the manuscript; or in the decision to publish the results.

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
