# Peer review of "Response Characteristics of Smoke Detection for Reduction of Unwanted Fire Alarms in Studio-Type Apartments"

_fire, doi:10.3390/fire6090362_

Round 1
Reviewer 1 Report
Recommondations:
Overall, it is a very interesting article that highlights a gap in the RSET calculation that is significantly important but has received little attention or investigation until now.
Section 2.1:
Line 82 - Basic terms that are not internationally known need to be clarified. The term “neighborhood facilities” is not known and used worldwide (Definition and explanation needed).
Line 87-89 - Are these the proportions of causes for false alarms across all building types or in relation to a specific building type. Please clearly define and differentiate.
Line 90 – 95 - It is slightly one-dimensional to think that the severity of the problem of false alarms of smoke detectors is determined by the number of installed products. At least a brief look at the method of detection should be taken and the reasons explained why smoke detectors must be more vulnerable to false alarms on the basis of their method of detection itself.
Line 95 – 102 - Please check again the terminology for the categorization of the false alarm types. There are already some publications which deal with the correct terminology on an international level.
Line 136 - Double listing of stir-fried vegetables. Can be deleted once.
Line 143-145 - Incorrect. These tests were carried out in order to develop a reproducible test basis and, if necessary, to define test specifications that must be fulfilled for the approval of smoke detectors.
Section 3.2
Line 185 - Lack of definition and description of analog and conventional smoke detectors. What’s the difference?
Section 4.2 Overall quite difficult to read due to the many shortcuts used
Line 273 – Estimated or real triggering time?
Line 286 – 288 - The fact that the positions of the individual smoke detectors differ massively in relation to the source of the “fire” was not taken into account in this interpretation. Please rework
Line 299 - Please discuss the relevance of an average value of activation times when the positions of the detectors are sometimes 5 meters or more from each other.
Line 328 – 335 - Why do we need a formula to calculate the activation time? As an average activation time?
Line 336 - Where does the formula come from. Derivation and/or Source is missing.
Line 336 ff - difficult to understand what was done here and why. please clarify
Line 356 -358 – Please explain Time SD-2 more detailed. Why calculating activation time of the smoke detectors 2 minutes after starting the experiment?
Section 4.3
Line 419 – 423 - Not comprehensible. Please describe and explain in more detail what has been done here and what are the results of it.
Section 4.4
Line 454 - Independent of the position of the detector in relation to the fire source?
Line 464 - Meaningfulness? 4 ppm CO is practically nothing . Please deal with it critically
Reviewer 2 Report
Manuscript ID: fire-2566166
Title: Response Characteristics of Smoke Detection for Reduction of Unwanted Fire Alarms in Studio-type Apartments
This research delves into how conventional and analog smoke detection systems respond in studio-type apartments, aiming to minimize false alarms. A simulation was conducted in accordance with applicable domestic regulations, standards, statistical data, and experimental instances. The objective of the study is worth investigating and authors have presented good methodology. Results highlighted are important for understanding smoke alarms. Paper meets the standard of journal and I recommend for publication. However, I have some comments as below:
1. The results of the work should be compared with the results of other investigators and/or other methodologies, experimental results or even simulated results. This is needed to place this work in perspective with other work in the field and provide more credibility for the present results. It would be more robust if the study compared the dataset with multiple alternative methods or datasets to validate the accuracy and reliability of the findings with previous literature.
2. How do authors justify the validation of the current study and, please briefly discuss the different models used in the previous study and their advantages/disadvantages to justify the current work? Authors may consider adding the accuracy data for other models and showing comparison and how authors came to conclusion.
3. Please mention the accuracy and make of different sensors used in the study.
4. Also include the variations in results due to repetition and their range on the graphs.
5. How can this data be used to improve fire and smoke detection during evacuations as in following studies- https://www.sciencedirect.com/science/article/pii/S037971122100120X and failure of smoke alarms https://link.springer.com/article/10.1007/s10694-019-00898-6
6. I am also interested in knowing how the results can be applied to high rise buildings especially in the stairs. This is a serious problem and present results can be very helpful. Please discuss smoke transport in stairwells using following recent studies.
https://www.mdpi.com/2075-5309/13/1/83,
https://elib.spbstu.ru/dl/2/k19-114.pdf/en/info,
7. Are there any limitations or challenges to using connected data in different scenarios? The study itself may have limitations related to the specific location or context of the fire event analyzed and location of sensors. Please discuss.
Minor editing of English language required
Author Response
Response to Reviewer 2 Comments
Thank you for seeking assistance in improving the quality of our work. In response to reviewer’s comments, we have made amendments to the following documents.
Point 1: The results of the work should be compared with the results of other investigators and/or other methodologies, experimental results or even simulated results. This is needed to place this work in perspective with other work in the field and provide more credibility for the present results. It would be more robust if the study compared the dataset with multiple alternative methods or datasets to validate the accuracy and reliability of the findings with previous literature.
Response 1: In the Approach section, we described how to verify the reliability and accuracy of the study.
Point 2: How do authors justify the validation of the current study and, please briefly discuss the different models used in the previous study and their advantages/disadvantages to justify the current work? Authors may consider adding the accuracy data for other models and showing comparison and how authors came to conclusion.
Response 2: The advantages and disadvantages of the study results are additionally described in Line 491-494.
Point 3: Please mention the accuracy and make of different sensors used in the study.
Response 3: We modified the maximum and maker.
Point 4: Also include the variations in results due to repetition and their range on the graphs.
Response 4: Average values are shown for readability of the paper.
Since we repeated 3 times per case, we should plot 18 graphs on one graph.
Nonetheless, if it's good to add, let me know. We'll add it.
Point 5: How can this data be used to improve fire and smoke detection during evacuations as in following studies- https://www.sciencedirect.com/science/article/pii/S037971122100120X and failure of smoke alarms https://link.springer.com/article/10.1007/s10694-019-00898-6
Response 5: Through these research results, it is possible to predict the location where unwanted fire alarms due to cooking by-products occur in a studio-type apartment, and it can be used to install smoke detectors to reduce unwanted fire alarms. We added In line 492-495.
Point 6: I am also interested in knowing how the results can be applied to high rise buildings especially in the stairs. This is a serious problem and present results can be very helpful. Please discuss smoke transport in stairwells using following recent studies. https://www.mdpi.com/2075-5309/13/1/83, https://elib.spbstu.ru/dl/2/k19-114.pdf/en/info
Response 6: Stairwells and hallways are currently in the process of follow-up research.
Point 7: Are there any limitations or challenges to using connected data in different scenarios? The study itself may have limitations related to the specific location or context of the fire event analyzed and location of sensors. Please discuss.
Response 7: Section 4.4 mentions the limitations of this paper. In addition, it is a standard limited to unwanted fire alarm tests.

Reviewer 3 Report
the authors did not provide how the methods and materials were designed, implemented and executed. consider to describe the setup materials and experimental process.
Author Response
Thank you for asking for our help to improve the quality of our work. The following article has been revised to reflect reviewer comments.
Please check the attached document. Thank you

Reviewer 4 Report
General Comments:
The authors investigated the response characteristics of conventional smoke detectors and Analog smoke detectors by simulating the burning of food during cooking, and measured the optical density, PM, and carbon monoxide levels when smoke detectors were activated, to provide a reference for reducing false fire alarms in studio-type apartments. Overall, the research content is interest, and the explanations are reasonable.
More specific comments are listed below to improve this manuscript.
1. The author should improve the English language. There are so many cases of unusual, vague or unclear writing.
2. Please check through the manuscript and spell out full names of all abbreviations at the first use in your manuscript. For example, CO, CO2, etc.
3. Some references are so old, and new related references are not cited. For example, Journal of Thermal Analysis and Calorimetry, 2022, 47: 14143–14153; Journal of Cleaner Production, etc.
4. In line 56, this study only mentioned the false fire alarm scenario when the food was burning, and the illustration of the research goal should be more specific here.
5. In lines 189-190, are these units used in explanations for PM1.0, PM2.5, and PM10.0 correct?
6. In line 336, is the abbreviation OD in Eq. 1 related to ODM in line 184? The abbreviation should be given its full name when it first occurs, and the units of variables in Eq. 1 should be stated in the text part.
7. In line 358, what the mean of SDobs? It appears only once in the text part.
8. In section 4.2, the accuracy of the formulas used in calculating the activation times of smoke detectors was not verified.
9. In lines 419-423, the contents were quoted from the references [14][15], whether are these contents relate to subsequent research?
10. In lines 428-429, what is EG in Eq. 10, 11, EG is not stated anywhere in the full text.
11. The conclusions are so lengthy. Please refine to show the innovation of this manuscript.
The author should improve the English language. There are so many cases of unusual, vague or unclear writing.
Author Response

(The authors gave the same response as above.)

Round 2
Reviewer 2 Report
Point 2: How do authors justify the validation of the current study and, please briefly discuss the different models used in the previous study and their advantages/disadvantages to justify the current work? Authors may consider adding the accuracy data for other models and showing comparison and how authors came to conclusion.
Response 2: The advantages and disadvantages of the study results are additionally described in Line 491-494.
Reviewer follow up point: I don't see this comment answered at all in these lines 491-494. Please check again and discuss in detail.
Point 4: Also include the variations in results due to repetition and their range on the graphs.
Response 4: Average values are shown for readability of the paper.
Since we repeated 3 times per case, we should plot 18 graphs on one graph.
Nonetheless, if it's good to add, let me know. We'll add it.
Reviewer follow up point: I am not suggesting to add data for all tests. But there is a way to show maximum deviation in the results using error bars. Almost all the scientific studies show these error bars to highlight the deviation in the results.
no
Author Response
Dear Reviewer,
We hope this message finds you well. We wanted to extend our sincere gratitude for taking the time to review our paper. Your expertise and insights have been invaluable in improving the quality of our work.
In order to address your comments and suggestions efficiently, we have attached a file.
Should you have any further questions or require additional information, please do not hesitate to reach out to us.
Once again, thank you for your invaluable contribution to our work. We look forward to your final evaluation of the revised manuscript.
Warm regards.

Reviewer 4 Report
Most issues were properly addressed by the authors. However, the authors only described some test and calculated results, and did not deeply discuss results and explain their reasons. Enough discussion is needed. Or else, this manuscript is more like a research report.
Some minor revisions are needed
Author Response

(The authors gave the same response as above.)
